# The Effect of a Single Dose of Citrulline on the Physical Performance of Soccer-Specific Exercise in Adult Elite Soccer Players (A Pilot Randomized Double-Blind Trial)

**DOI:** 10.3390/nu14235036

**Published:** 2022-11-26

**Authors:** Eduard Bezuglov, Ryland Morgans, Artemii Lazarev, Evgeny Kalinin, Mikhail Butovsky, Evgeny Savin, Eduard Tzgoev, Bekzhan Pirmakhanov, Anton Emanov, Andrey Zholinsky, Oleg Talibov

**Affiliations:** 1Department of Sports Medicine and Medical Rehabilitation, Sechenov First Moscow State Medical University of the Ministry of Health of the Russia, 119991 Moscow, Russia; 2High Performance Sports Laboratory, Moscow Witte University, 115432 Moscow, Russia; 3The Academy of the Russian Football Union, 115172 Moscow, Russia; 4PFC CSKA, 125167 Moscow, Russia; 5Department of Internal Medicine, Mount Sinai Hospital, Chicago, IL 60608, USA; 6Sports Adaptology Lab, Moscow Institute of Physics and Technology (National Research University), 141701 Dolgoprudniy, Russia; 7Football Club “Rubin”, 420036 Kazan, Russia; 8Department Neurology and Rehabilitation, Kazan State Medical University, 420012 Kazan, Russia; 9Higher School of Economics, 101000 Moscow, Russia; 10Department of Epidemiology, Biostatistics and Evidence-Based Medicine, Faculty of Medicine and Health Care, Al-Farabi Kazakh National University, Almaty 050040, Kazakhstan; 11Football Club Kairat, Almaty 050054, Kazakhstan; 12“Smart Recovery” Sports Medicine Clinic LLC, 121552 Moscow, Russia; 13Federal Research and Clinical Center of Sports Medicine and Rehabilitation, Federal Medical Biological Agency, 121059 Moscow, Russia; 14Department of Internal Medicine, Clinical Pharmacology and Emergency Medicine, Moscow State University of Medicine and Dentistry, 127473 Moscow, Russia

**Keywords:** citrulline, elite soccer players, soccer, performance, fatigue, lactate, creatine kinase, rating of perceived exertion

## Abstract

Purpose: The purpose of this study was to evaluate the effect of a single intake of citrulline at 3 g and 6 g doses in adult elite soccer players performing sport-specific exercise. Materials and Methods: This randomized double-blind placebo-controlled study analyzed 18 soccer players from the top divisions of three European countries. Participants were randomized into three groups of six each and performed a field-based soccer-specific test for 18 min. Comparative analysis of heart rate, fatigue and post-exercise recovery was conducted. Results: There were no statistically significant differences in most of the analyzed parameters, nor at any of the time points for lactate concentration. Players’ RPE exercise test score did not reveal any differences. Conclusions: Neither a single intake of 3 g nor of 6 g of citrulline malate affected physical performance, subjective feelings of fatigue or post-exercise recovery in adult elite soccer players who performed a soccer-specific test.

## 1. Introduction

L-Citrulline is an essential amino acid found naturally in the body. Despite conflicting data on the effect of citrulline on various aspects of performance, this substance is widely used in sports, and notably, there is certain interest in its application in elite sports [1].

Citrulline is an endogenous precursor of L-arginine, which is the main substrate for NO-synthase of nitric oxide (NO) in the NO-synthase (NOS) pathway [1]. The potential positive effect of citrulline on various aspects of physical performance has been revealed in previous research [2,3,4]. Citrulline has been shown to improve physical performance through improvement of metabolism of ammonia and lactic acid, thereby improving oxygen delivery by increasing vasodilation and production of adenosine triphosphate, as well as increasing intermediate products of the Krebs cycle [5]. Furthermore, citrulline intake has been shown to cause a significant increase in rate of oxidative production of adenosine triphosphate during exercise and to accelerate recovery of phosphocreatine after exercise [2].

According to Jagim et al. (2019), around 70% of the most popular pre-workout drink complexes contain citrulline. These complexes, often used by both sports amateurs and professionals, have an average citrulline dosage of 4 g ± 2.5 g [3]. Therefore, citrulline can be considered one of the most frequently used pharmacological substances in athletes of different levels, despite the fact that there is still no convincing evidence of its effectiveness [6].

However, there is some evidence to suggest effectiveness of citrulline in strength training and aerobic exercise, as well as in rate of post-exercise recovery. In a meta-analysis conducted by Rhim et al. (2020), citrulline supplementation was shown to significantly reduce post-workout rating of perceived exertion (RPE) and muscle soreness without affecting lactate levels [4]. Further systematic reviews by Gonzalez and Trexler (2020) and by Trexler et al. (2019) suggested that there is evidence to support the notion that citrulline supplementation can improve physical performance and post-exercise recovery [7,8]. In contrast, other studies did not demonstrate a positive effect on physical performance and post-exercise recovery [9,10,11]. Therefore, the currently available data on the impact of citrulline on physical performance and post-exercise recovery can be considered controversial.

Noteworthily, a significant limitation of the existing literature is that the majority of participants were physically healthy individuals and not professional athletes, and the exercise protocols utilized in these studies were not specific to any particular sport. Additionally, it should also be considered that ergogenic response of citrulline or arginine supplementation may have depended on basic fitness level of the subjects. Studies in untrained or moderately trained healthy subjects have shown that NO donors can improve tolerance of aerobic and anaerobic exercise [12]. However, when highly skilled subjects were examined, no positive impact on performance was observed [13]. Therefore, research investigating the effect of different citrulline doses on physical performance and post-exercise recovery in professional adult soccer players would be of significant interest to scientific and applied practitioners.

The aim of this trial was to examine the effect of a single dose of 3 g or 6 g of citrulline malate on physical performance, subjective perception of fatigue and post-exercise recovery in adult professional soccer players following completion of a soccer-specific test.

## 2. Materials and Methods

### 2.1. Participants

This study examined 18 elite male outfield soccer players from the top divisions of three European countries. Players were classified by position and grouped accordingly: defenders, *n* = 8; midfielders, *n* = 8; and forwards, *n* = 2. All data evolved as a result of employment in which players were routinely monitored over the course of the competitive season. This study was performed in accordance with Helsinki Declaration principles. Ethical approval was granted by the local Ethics Committee of Sechenov University (N05-21 dated 10.3.21). To ensure confidentiality, all data were anonymized before analysis. Participants were fully familiarized with the experimental procedures within this study due to the regular testing protocols implemented as part of the clubs’ performance-monitoring strategies. During this study, players were instructed to maintain normal daily food and water intake, and no additional dietary interventions were undertaken.

Participant age and anthropometric data are presented in Table 1.

Citrulline malate (Stimol) manufactured by Biocadex (France) was utilized in this study. The content was 1 g of citrulline malate in 10 mL of orange-flavored solution. A single oral intake of 3 g or 6 g of citrulline malate were administered 45 min prior to the start of exercise. This dosage has been previously validated in handball athletes [14]. A double-blind study design was performed in three parallel groups: two groups received a standard or double dose of citrulline malate, while the third group consumed a placebo of orange juice that was similar to the citrulline solution in color and taste. An independent researcher (team nutritionist), who did not participate in this research study, was responsible for placebo preparation and administration according to the randomized list.

### 2.2. Procedure

All participants were randomly assigned to one of three groups: Group 1 received a 6 g dose of citrulline malate, Group 2 received a 3 g dose of citrulline malate and Group 3 was administered a placebo solution. In order to mask the sour taste of citrulline malate for Groups 1 and 2, dosages were diluted in 250 mL of freshly squeezed orange juice one hour before consumption. The citrulline malate dose was consumed by participants 2 h after a standard breakfast and 45 min before the start of the standardized warm-up. Soccer-specific physical activities were then performed.

Randomization was conducted using the R 4.0.2 “randomizeR” package. The resulting randomization codes were relayed to the team’s nutritionist, who prepared and labeled all drinks. Prior to the soccer-specific tests, citrulline malate was pre-dispensed into prepared labeled bottles that correlated with the randomization list. During testing, the identity of the substance received by each player remained anonymous. Test results were recorded for later analysis by blinded-study staff.

### 2.3. Soccer-Specific Exercise Protocol

Before soccer-specific protocol began, a standardized warm-up lasting 20 min in duration was performed under supervision of a qualified fitness coach. The soccer-specific protocol consisted of two series of 9 min performed on a natural-turf soccer field, with 6 min recovery between each series. At the start of each series, the players were grouped and positioned on the penalty area, as shown in Figure 1. Each group was positioned opposite the other. In the center of the field, 5 cones were placed at a distance of 2 m apart. The distance from the penalty area to the first cone was 25 m (see Figure 1).

On a signal from the coach, players started running in a straight line; when players arrived at the cones, each dribbled the ball through the cones, continued to run and, at 5 m before the penalty box (marked by a cone), passed the ball to the next player. The duration of each run was 10 s. The passive rest interval between each run was 20 s. The total duration of one series was 540 s. The rest interval between series was 360 s.

### 2.4. Equipment Used and Analyzed Parameters

Physical performance data were collected and analyzed for the soccer-specific exercise using a global positioning satellite (GPS) tracking system (WIMU PRO). The WIMU PRO device (Realtrack Systems, Almería, Spain) is comprised of different sensors, including four accelerometers (1000 Hz), three gyroscopes (1000 Hz), a barometer (100 Hz), a magnetometer (100 Hz), a global navigation satellite system chip (GNSS, M = 8.96, SD = 1.56) and a UWB chip. It has been previously validated to quantify soccer demands. Location technology and accelerometers were used to directly obtain distance covered and speed, while the other variables analyzed were manually calculated or obtained from other variables [15]. The GPS signal during data collection was 10 Hz (10 frames per second, or 1 data point every 100 ms), so values higher than 3 m/s^2^ (for 100 ms) were quantified as acceleration. Specially designed vests were used to hold the devices. Each vest was worn on the player’s upper torso and anatomically adjusted to each player, as previously described (Malone et al., 2017). Heart rate was also measured, using a Garmin chest strap synchronized with the WIMU system [15,16,17,18].

Using this system, the following physical metrics were evaluated: average heart rate during exercise; maximum heart rate; heart-rate recovery at 1, 3 and 5 min after exercise; maximum and average speed during exercise; and distance covered at different speed thresholds. Motor actions were classified according to specific speed thresholds: walking at <7.2 km/h, jogging at 7.2–14.4 km/h, running at 14.4–19.8 km/h, high-intensity running at 19.8–25.2 km/h and sprinting at >25.2 km/h. These thresholds have been previously validated [19,20].

All finger capillary blood samples were collected immediately before the soccer-specific test, 3 min after both series and 30 min after the second series of exercise, for lactate analysis. Finger capillary samples were collected from a standing position using a sterile lancet (Roche, Mannheim, Germany) in combination with a spring-loaded AccuChek lancet device (Roche, Mannheim, Germany). Lactate samples were analyzed immediately after collection via a Lactate Scout device (Leipzig, Germany) (measurement range: 0.5–25.0 mmol/L; measurement time: 10 s). Capillary blood analyzed using this method displayed intra-assay reliability of 3–8% coefficient of variation.

Finger capillary blood was also taken on the morning of the test day prior to breakfast as well as 18 h after the start of testing from all participants. The selection of this time range was based on extensive research stating that maximum concentration of creatine kinase (CK) is observed 12–24 h after the end of exercise [21,22]. Every effort was made to control a range of factors that had the potential to impact CK concentration. These included time of day and immediate nutritional intake prior to sample collection. Other potentially important variables, such as nutrition and activity not associated with official training sessions, could not be rigidly controlled during data collection due to the nature of this investigation. These factors were, however, unlikely to impact the data to any large extent as a consequence of similarity in individual players’ routines. Creatine kinase samples were analyzed immediately after collection via spectrophotometry using a commercially available reagent kit (Reflotron^®^ Systems, Roche, Mannheim, Germany). Capillary blood analyzed using this method displayed intra-assay reliability of <3% coefficient of variation.

Ten minutes after completion of the second series, each subject provided information on rating of perceived exertion (RPE): a reliable, subjective measure of severity of exercise load that apparently is not influenced by time elapsed after training [23]. As previously validated, RPE is a generally accepted method of monitoring training load in athletes involved in many different sports, and is based on the category scale of 0–10 RPE (BORG-CR10), developed by Borg [24,25]. On this scale, a score of 0 corresponds to no fatigue and a score of 10 indicates severe fatigue.

## 3. Statistical Analysis

Statistical analysis was performed using statistical package GraphPad Prism 9.0 for Mac OS X. Description of quantitative data is provided as mean and standard deviation (M ± SD). To establish statistical significance of differences between groups, the Kruskall–Wallis test (a non-parametric analogue of ANOVA) was used. Speed differences within groups during the first and second series were assessed with a Wilcoxon test. Differences were considered statistically significant at *p* < 0.05. Differences with *p* < 0.1 were further analyzed, considering the limited sample size. Exact *p*-values were calculated for all comparisons. Initial statistical analysis was performed by the blinded-study staff member. Unblinding was performed after the full data set was analyzed.

## 4. Results

Prior to the first series of exercise, all participants demonstrated increased heart rate due to the 5 min standardized warm-up. As is evident from Table 2, all players showed an increase in heart rate at peak intensity of the exercise, followed by significant recovery within 5 min. In analysis of heart rate in all groups and across all time points, the only statistically significant difference (*p* = 0.03) was revealed at the third minute of recovery after the second series of exercise was performed.

No statistically significant differences in lactate concentration were revealed at any analyzed time point. A statistically significant increase in lactate concentration (*p* < 0.001) at 3 min post-test was observed in each group when comparing lactate concentration at the start of the test in all groups. However, no inter-group differences were observed. Thirty minutes after completion of the second series of exercise, all three groups showed a significant decrease in lactate concentration compared to pre-exercise values (*p* < 0.001), although no inter-group differences were reported. Lactate-concentration changes are shown in Table 3 and Figure 2.

Concentration of CK (Table 3, Figure 3) across all groups revealed an increase 18–20 h after the test exercise as compared to the basic level (*p* < 0.001). In comparison of differences between groups, no significant differences were found; however, lower CK values were reported in Group 1 (*p* = 0.08). Players’ RPE exercise test score did not reveal any differences (*p* = 0.51). In Group 1, one player scored at the intensity at 10, two players scored at 9 and three players scored at 8. In Group 2, two players rated the intensity at 9 and 8, respectively.

In Group 3, two players reported a score of 9 while the remaining four players scored at 8 (Figure 4).

Table 3 represents data for heart-rate changes. The only statistically significant difference was revealed in the third minute after the first exercise series in the placebo group.

Table 4 presents data that characterizes speed parameters for all participants recorded during the test exercise. In comparison of values across groups, the average and maximum speeds achieved in the first and second series of exercise showed no statistically significant differences. In comparison of these parameters within the groups for the first and second series of test exercise, no statistically significant differences were found (Wilcoxon test, *p* > 0.05 for all three groups). Additionally, when comparing players’ time intervals, speeds of 14.4–19.8, 19.8–25.2 and more than 25.2 km/h were achieved. With the exception of time of maximum acceleration in the first series of exercise, no differences were reported. The only difference reported was in sprint duration for one participant in Group 2 and one participant in Group 3. When comparing differences within each group, no statistically significant differences were evident between the first and second series of tests (Wilcoxon test, *p* > 0.05).

## 5. Discussion

The present pilot trial reported no data to suggest that a single intake of citrulline malate in a dose of 3 g or 6 g, 1 h prior to a soccer-specific test at maximum intensity, has a positive impact on physical activity, physical performance, subjective feelings of fatigue or post-exercise recovery in adult professional soccer players. Neither of these doses produced any physical improvements in relation to locomotor activity at different speed thresholds, average or maximum heart rate during exercise, heart-rate recovery at different recovery time periods (RPEs) or lactate and CK concentrations. This study, to our knowledge, is the first to examine the effect of citrulline malate on the physical-activity profile of adult professional soccer players during sport-specific exercise, and therefore, comparisons with the existing literature is difficult. Furthermore, limited available literature employed a randomized approach to examine the impact of a single citrulline malate dose or course of dosages on RPE, heart rate and concentrations of lactate and CK.

However, our findings are supported by the limited existing literature. For example, 10 well-trained male cyclists consumed a single 12 g dose of citrulline 60 min prior to repetitive cycling sprints, and although elevated exercising heart rate was found no positive effect on RPE was reported [9]. Additionally, in several subject groups from the general population to amateur and well-trained athletes that consumed 8 g of citrulline malate prior to exercise, no positive changes in RPE, muscle fatigue, lactate concentrations [10,11] and heart rate and lactate concentrations [26] were reported. However, an associated increase in cardiovascular dynamics and resistance exercise performance was observed in 14 well-trained men [26]. Notably, in analysis of heart-rate data in all groups and across all time points of our participants, the only statistically significant difference (*p* = 0.03) was revealed at the third minute of recovery after the second series of exercise was performed. However, given that this was noted in the placebo group and given the absence of confirmatory changes in other time points, this effect was attributed by the authors to possible consequences of multiple testing or to the physical characteristics of the three participants in this group. These participants also demonstrated lower peak heart rate and faster heart-rate recovery after the first series of the soccer-specific test. During the second series of the soccer-specific test, the highlighted differences were already absent. The average heart rate during our soccer-specific test protocol was also comparable to previously published results during competitive soccer matches [27].

Although the effect of citrulline ingestion on markers of strength was not examined in our study, others have evaluated the effect of a single dose of 8 g of citrulline and found contrasting results. In a study of 19 young participants, authors concluded that citrulline did not positively contribute to the effectiveness of a strength-training program, and lactate and CK concentration were unchanged; however, muscle soreness was significantly reduced [28]. Another study, which investigated 15 young women, reported that a single 8 g dose of citrulline 60 min prior to strength-training exercises performed to failure, resulted in greater effectiveness of upper and lower body resistance exercise and a decrease in RPE, although no positive effect on heart rate was found [29]. Therefore, due to the conflicting data available, further research is warranted to examine effects of citrulline on strength programs in a soccer-specific population.

Notably, in a recent study that administered a different consumption protocol (6 g of citrulline daily for 7 days; the final intake was administered 120 min prior to the test), positive correlations in heart rate, RPE and average power output during cycling and a 40 km race time were reported in nine trained male cyclists [30]. Additionally, although with different dosing protocols, further studies examined the effect of ingestion of a course of citrulline by professional athletes. In 72 high-performing endurance athletes who consumed a daily dose of 3 to 6 g of citrulline for 13 days, lactate concentration and daily RPE after exercise were measured, and data from this cohort indicated a positive effect of citrulline on lactate recovery rate and RPE [31]. Similarly, in handball players, blood lactate concentrations significantly decreased after consumption of 1 g of citrulline three times a day for one month, especially immediately after training [14]. However, these findings contradicted our results with a similar athletic population (well-trained professional soccer players), and this may partly be explained by participant heterogeneity: notably age, anthropometry, fitness level and citrulline consumption protocols employed. In the majority of randomized trials, participants were amateur athletes or healthy, physically active subjects, aged 18 to 51 years, from the general population: not professional athletes. Furthermore, to our knowledge, only two studies have examined the effect of citrulline in team sports, such as in Turkish handball players from the first league or in university basketball and soccer teams [14]. Most studies have assessed the effect of a single dose of citrulline on various aspects of physical performance, cardiovascular function and post-exercise recovery. In these studies, the dose of citrulline has ranged from 1 to 12 g at 40–120 min prior to exercise, though the most commonly employed single dose has been 6 to 8 g. Arguably, the citrulline malate protocol used in our study is one of the most commonly practiced in terms of dosage, frequency and time of administration prior to exercise. Our study did not observe a dose-dependent effect in relation to any analyzed parameters. However, the exercise protocol utilized can be considered the most “sport-specific” in relation to soccer actions: running at varying intensities; number of accelerations, decelerations and changes of direction; and dribbling at high-speed. Furthermore, the protocol was also conducted on a natural field, in soccer attire and with participants wearing appropriate soccer footwear.

The limitations of our study include a relatively small number of participants. The main reason for this small sample size was lack of top-level soccer players available to participate by receiving the required dose and completing the physical test in controlled experimental conditions. Other notable limitations include absence of hormonal profile analysis (i.e., cortisol- and testosterone-level changes).

The lack of available data highlighting the positive impact of a single dose or course of citrulline doses in professional athletes warrants further investigation. Moreover, previous studies involving endurance athletes and handball players have shown a positive effect on lactate concentration after exercise, though the supporting literature is scant and warrants further investigation. Specific consumption protocols still need further examination to ensure that practical application is based on scientific findings. Furthermore, our study did not assess severity of delayed onset of muscle soreness (DOMS) at different recovery time periods after exercise. Given the proven negative effect of DOMS on physical performance and given the previous literature that has shown, among other things, a positive effect of citrulline on severity of DOMS, it would be interesting to evaluate this notion in this specific group of soccer players. Therefore, it is possible to suggest that citrulline intake provides greater benefits to athletes with a chronic rather than an acute intake protocol. Furthermore, studies evaluating potential benefits of long-term citrulline supplementation are of great practical and scientific interest.

## 6. Conclusions

Neither a single dose of 3 g nor a single dose of 6 g of citrulline malate positively affected physical activity, physical performance, subjective feelings of fatigue or post-exercise recovery in adult elite soccer players who performed a soccer-specific test.

## Figures and Tables

**Figure 1 nutrients-14-05036-f001:**
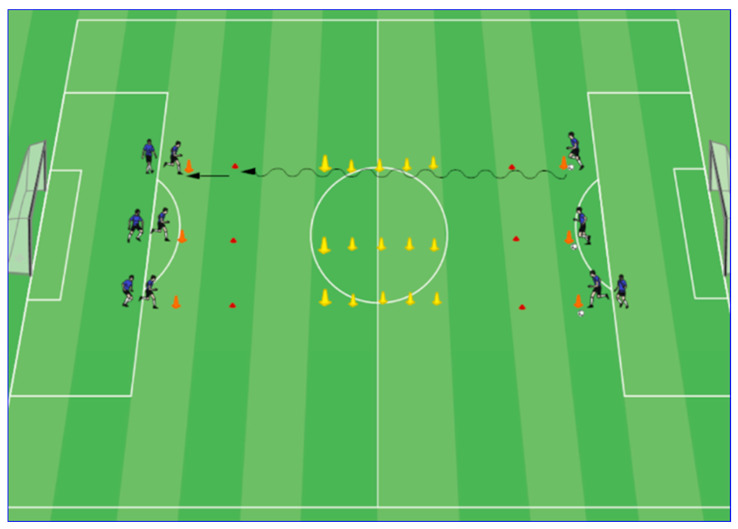
Schematic diagram of the soccer-specific exercise protocol of dribbling with a ball.

**Figure 2 nutrients-14-05036-f002:**
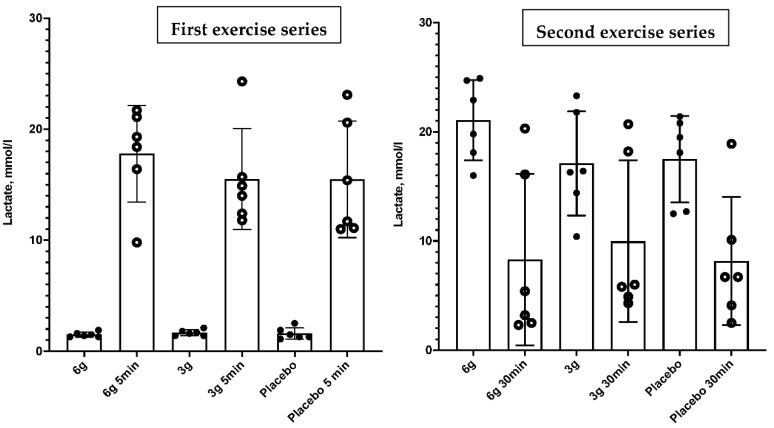
Blood-lactate changes before and after the field-based soccer-specific test.

**Figure 3 nutrients-14-05036-f003:**
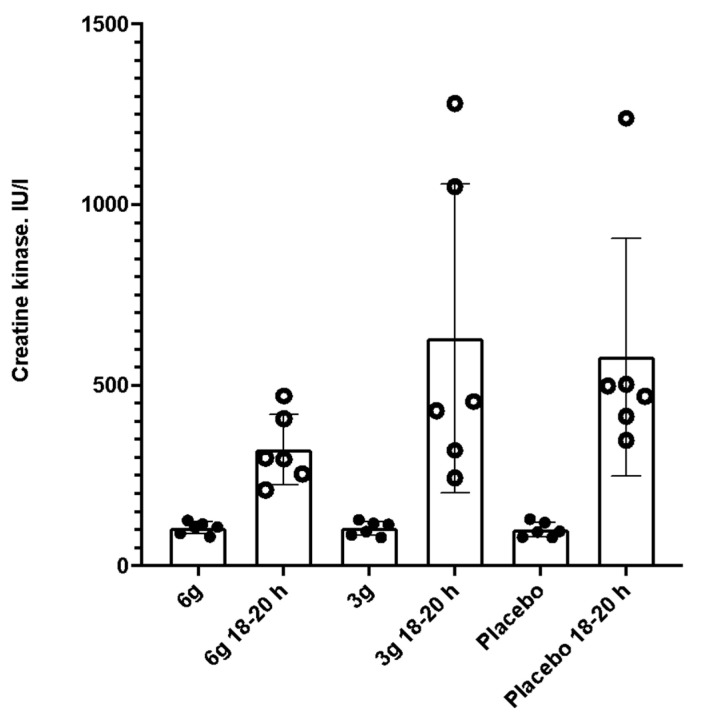
Creatine-kinase changes before and after the field-based soccer-specific test.

**Figure 4 nutrients-14-05036-f004:**
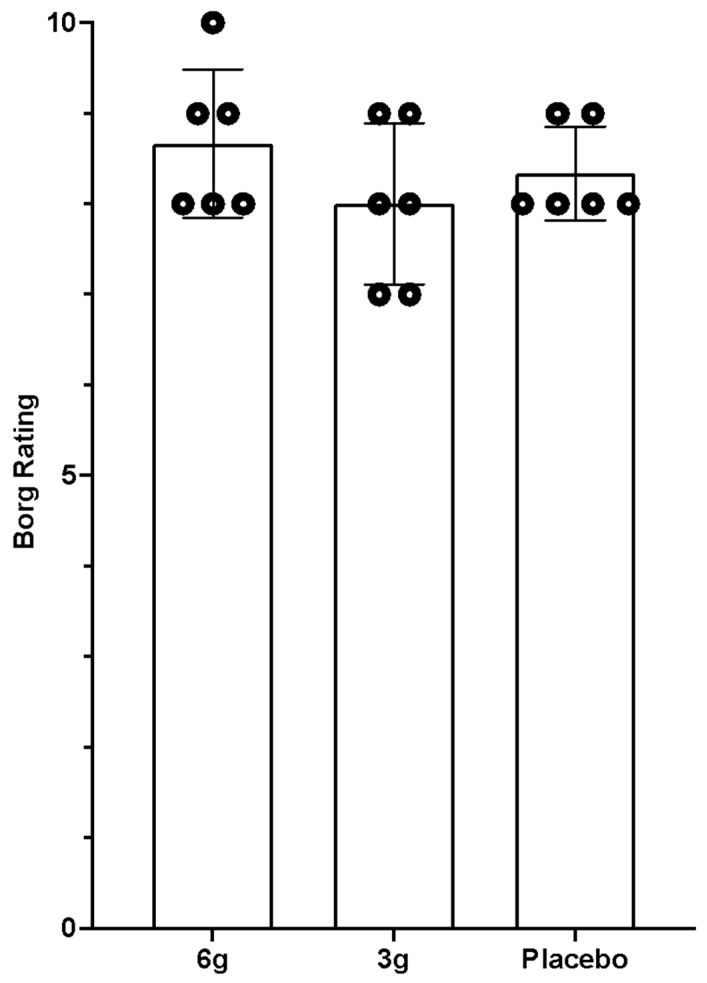
Rating of perceived exertion (Borg’s scale).

**Table 1 nutrients-14-05036-t001:** Participant age and anthropometry.

	Citrulline 6 g (*n* = 6)	Citrulline 3 g (*n* = 6)	Placebo (*n* = 6)	*p* Value
Age	24 ± 2	26 ± 5	28 ± 4	0.33
Weight (kg)	83 ± 8	77 ± 8	84 ± 9	0.25
Height (cm)	185 ± 5	184 ± 5	188 ± 8	0.56
BMI (kg/m^2^)	24 ± 2	23 ± 1	24 ± 1	0.20

**Table 2 nutrients-14-05036-t002:** Exercise-test heart-rate variables for all groups.

	Citrulline 6 g (*n* = 6)	Citrulline 3 g (*n* = 6)	Placebo (*n* = 6)	*p* Value
Exercise 1				
HR Baseline (M ± SD)	101.5 ± 16.5	102.3 ± 5.3	99.83 ± 6.6	0.83
HR Average (M ± SD)	167.8 ± 9.4	170.2 ± 6.8	162 ± 8.6	0.29
HR Max (M ± SD)	175.7 ± 8.9	180.3 ± 7.0	170.7 ± 6.8	0.13
HR Recovery 1st min (M ± SD)	150.3 ± 13.1	149.7 ± 5.0	136.7 ± 9.5	0.05
HR Recovery 3rd min (M ± SD)	121.3 ± 15.5	123.5 ± 5.2	110.7 ± 5.5	**0.0272**
HR Recovery 5th min (M ± SD)	118.7 ± 15.3	121 ± 5.8	108.8 ± 5.4	0.09
Exercise 2				
HR Baseline (M ± SD)	120.5 ± 13.2	120.3 ± 7.3	112.8 ± 11.8	0.4
HR Average (M ± SD)	170 ± 8.9	172.8 ± 6.7	163.7 ± 7.8	0.21
HR Max (M ± SD)	177.2 ± 8.7	182.8 ± 6.3	173.5 ± 7.5	0.14
HR Recovery 1st min (M ± SD)	152.7 ± 13.7	150 ± 8.3	141.2 ± 5.9	0.21
HR Recovery 3rd min (M ± SD)	120.7 ± 12.7	122.7 ± 7.6	114 ± 6.9	0.24
HR Recovery 5th min (M ± SD)	117.5 ± 10.3	121 ± 8.3	111.8 ± 5.9	0.23

**Table 3 nutrients-14-05036-t003:** Biochemical parameters for all groups before and after the field-based soccer-specific test.

	Citrulline 6 g (*n* = 6)	Citrulline 3 g (*n* = 6)	Placebo (*n* = 6)	*p* Value
Lactate Measure (mmol/L), Exercise 1				
Baseline (M ± SD)	1.5 ± 0.23	1.667 ± 0.27	1.6 ± 0.52	0.52
Recovery 3rd min (M ± SD)	17.78 ± 4.35	15.52 ± 4.55	15.48 ± 5.25	0.59
Lactate Measure (mmol/L), Exercise 2				
Lactate Recovery 3rd min (M ± SD)	21.07 ± 3.67	17.1 ± 4.77	17.5 ± 3.96	0.27
Lactate Recovery 30th min (M ± SD)	8.3 ± 7.86	9.983 ± 7.4	8.167 ± 5.86	0.61
Creatine Kinase (IU/L)				
Baseline (M ± SD)	105.7 ± 16.6	104.55 ± 19.6	100.8 ± 20.9	0.96
18–20 h After Exercise (M ± SD)	323 ± 97.9	629.8 ± 427.7	578.7 ± 329.3	0.08

**Table 4 nutrients-14-05036-t004:** Exercise-test GPS metrics for all groups.

	Citrulline 6 g (*n* = 6)	Citrulline 3 g (*n* = 6)	Placebo (*n* = 6)	*p* Value
Exercise 1				
Average Speed (M ± SD)	10 ± 0.58	11 ± 0.58	10 ± 1.1	0.45
Maximal Speed (M ± SD)	25 ± 0.8	27 ± 104	27 ± 2.4	0.07
Total Distance				
<7.2 km/h (Walking)				
7.2–14.4 km/h (Jogging)				
Speed 14.4–19.8 km/h, s (M ± SD) (Running)	413 ± 62	397 ± 76	383 ± 78	0.73
Speed 19.8–25.2 km/h HSR), s (M± SD)	192 ± 75	261± 101	236 ± 100	0.42
Speed >25.2 km/h, s (M ± SD) (Sprinting)	0.83 ± 0.98	3.8 ± 3.8	5.3 ± 4.0	**0.036**
Exercise 2				
Average Speed (M ± SD)	10 ± 0.24	10 ± 0.69	10 ± 0.91	0.52
Maximal Speed (M ± SD)	26 ± 1.5	25 ± 2.4	26 ± 1.3	0.48
Speed 14.4–19.8 km/h, s (M ± SD)	418 ± 47	439 ± 61	435 ± 84	0.73
Speed 19.8–25.2 km/h, s (M ± SD)	116 ± 68	205 ± 72	184 ± 78	0.1735
Speed >25.2 km/h, s (M ± SD)	1.2 ± 2.4	3.3 ± 4.3	1.2 ± 1.3	0.61

## Data Availability

All data generated or analyzed during this study are included in this article. Further inquiries can be directed to the corresponding authors.

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
