# Peer review of "The Effect of a Single Dose of Citrulline on the Physical Performance of Soccer-Specific Exercise in Adult Elite Soccer Players (A Pilot Randomized Double-Blind Trial)"

_nutrients, 2022, doi:10.3390/nu14235036_

Round 1
Reviewer 1 Report
The authors have conducted an ecologically valid study investigating the effects of 2 doses of citrulline on football-specific performance. The authors acknowledge that the scientific evidence for performance benefits from citrulline ingestion are limited however they also demonstrate that citrulline ingestion for performance in a practical context is prevalent. They have performed a study to examine whether there are any meaningful effects in professional football players. The study design is generally fine (although I have some comments to follow) and the results and conclusions (no performance effects following ingestion) appear robust. I have some comments on the study and also will list some grammatical/typo errors. I also feel that the manuscript would benefit from proof reading by a native English speaker.
General comments:
I would like clarification on the football-specific running. The manuscript states that players ran at 115% of individual VO2max yet there are no individual data for VO2max presented and it appears that all players ran at the same absolute speed during the activity. This would appear to contradict the statement that they were running at individualised/relative speeds. (Unless all 18 players had exactly the same VO2max values!).
(lines 150-155) Further to the above, if players are all running at the same absolute speeds as stated then I do not understand why you would expect to see any differences in the GPS data for running speed distances, thresholds etc.
Finally (and you sort of address this in the limitations) why did you not have a cross-over design? This would eliminate any issues in inter-group differences (eg lines 283-287 when you discuss differences in HR).
Specific points:
Line 72 - seems t be missing a word (perhaps "tests")
Line 82 should read "subjective perception" not "subjective perceiving"
line 92 - "...was gram 10ml of orange..." seems like some words are missing
line 100-101 - perhaps re-write to "An independent researcher who did not participate..."
Line 123 - I would just write this as "cones" not cone-stands - same for line 125 where you call them cone-posts
Tables - for your data you should change the comma to a period to denote the decimals eg 8.32 not 8,32
Table 4 states xercise 1 not Exercise 1
Line 294 "...a study by in 15..." (delete either by or in)
Thanks
Author Response
Comments and Suggestions for Authors
The authors have conducted an ecologically valid study investigating the effects of 2 doses of citrulline on football-specific performance. The authors acknowledge that the scientific evidence for performance benefits from citrulline ingestion are limited however they also demonstrate that citrulline ingestion for performance in a practical context is prevalent. They have performed a study to examine whether there are any meaningful effects in professional football players. The study design is generally fine (although I have some comments to follow) and the results and conclusions (no performance effects following ingestion) appear robust. I have some comments on the study and also will list some grammatical/typo errors. I also feel that the manuscript would benefit from proof reading by a native English speaker.
General comments:
I would like clarification on the football-specific running. The manuscript states that players ran at 115% of individual VO2max yet there are no individual data for VO2max presented and it appears that all players ran at the same absolute speed during the activity. This would appear to contradict the statement that they were running at individualized/relative speeds. (Unless all 18 players had exactly the same VO2max values!).
This figure of 115% of individual VO2max used only to determine participants whose load was above required values. This value did not have an impact on the exercise. Thus, manuscript text has been corrected.
(Lines 150-155) Further to the above, if players are all running at the same absolute speeds as stated then I do not understand why you would expect to see any differences in the GPS data for running speed distances, thresholds etc.
This data provided to show that stereotype performance of field-based soccer-specific test gives similar load for different group participants.
Finally (and you sort of address this in the limitations) why did you not have a cross-over design? This would eliminate any issues in inter-group differences (eg lines 283-287 when you discuss differences in HR).
In general cross-over design is a good choice for studies when a subject condition does not significantly change between study periods (i.e. within-subject variation is lower than inter-subject variation). In the trial that includes limited amount of professional sportsmen within-subject variation could be as high as inter-subject because of different periods (before or after competition etc.) Moreover possibility of trauma should be taken into account especially in the limited sample size.
Specific points:
Line 72 - seems to be missing a word (perhaps "tests") I didn’t find this mistake
Line 82 should read "subjective perception" not "subjective perceiving" - Corrected
line 92 - "...was gram 10ml of orange..." seems like some words are missing 1 g in 10 ml - corrected
line 100-101 - perhaps re-write to "An independent researcher who did not participate..." Corrected
Line 123 - I would just write this as "cones" not cone-stands - same for line 125 where you call them cone-posts - Corrected
Tables - for your data you should change the comma to a period to denote the decimals eg 8.32 not 8,32 - Corrected
Table 4 states exercise 1 not Exercise 1 - Corrected
Line 294 "...a study by in 15..." (delete either by or in) - Corrected

Reviewer 2 Report
The aim of this study was to evaluate the efficacy of a single intake of citrulline at various doses in adult 26 professional soccer players performing sport-specific exercise at maximum intensity. The paper is interesting, and the methodology is well designed and performed. However, the paper needs proof-reading in English, as well as the authors need to considerably improve the introduction and discussion before the paper can be accepted for publication.
Abstract
The abstract is very short, especially the results section. The authors should try to describe the main results providing some statistical significance and more information abouth the physical capacities evaluated.
Introduction
Page 1 Lines 40-43. No reference is provided for this information. Please provide a at least a scientific and recent reference to support this affirmation.
Page 2 Lines 53-57. Authors should provide a reference for the study of Jagim et al (2019).
Page 2 Lines 59. “There is however evidence”. Please review the grammar of this sentence.
Page 2 Lines 66 to 68. If there is not empirical data “that” highlight a positive effect on physical performance, why is it controversial? Controversial is when some studies found effects and others did not find effects. Please the authors should rephrase this paragraph to justify the necessity of performing this study.
Page 2 Lines 70 to 80. I do not understand the rationale of this paragraph that describes how the effects of citrulline are not found neither for untrained, moderately trained and professional athletes. This paragraph should be rewritten to describe with some evidence why it seems that citrulline does not have effects of all type of athletes.
The introduction lacks of hypothesis. I think it would be very interesting to see what the authors predicted before performing the study. Do you think that citrulline was going to have certain effects?
Methods
Sample
Why is this study done in soccer players? Is there any special reason? Please justify why this sample.
Procedure
Authors must explain and justify why they used this specific exercise and no other. Is this exercise representative of the real efforts of soccer? Has this exercise been used in previous studies? in which studies? Please describe this.
Results
Results are well presented
Discussion
The discussion needs English proof-reading and a more elaborated interpretation of the results.
From the line 290 to 326 the authors cite and explain multiple studies but they should have been create relationship between them and summarize their findings, rather than explaining all of them. Please review and rephrase considerably this part of the discussion.
An example of proof-reading is in Page 8 Line 269. Please rephrase the sentence “it´s found”, this is not a typical expression in academic and formal English.
Major revision is needed before acceptacion of the paper.
Author Response
Comments and Suggestions for Authors
The aim of this study was to evaluate the efficacy of a single intake of citrulline at various doses in adult 26 professional soccer players performing sport-specific exercise at maximum intensity. The paper is interesting, and the methodology is well designed and performed. However, the paper needs proof-reading in English, as well as the authors need to considerably improve the introduction and discussion before the paper can be accepted for publication.
Abstract
The abstract is very short, especially the results section. The authors should try to describe the main results providing some statistical significance and more information about the physical capacities evaluated.
Introduction
Page 1 Lines 40-43. No reference is provided for this information. Please provide a at least a scientific and recent reference to support this affirmation. Reference (1) added
Page 2 Lines 53-57. Authors should provide a reference for the study of Jagim et al (2019). Reference (3) is added in text. Jagim, A. R., Harty, P. S., & Camic, C. L. (2019). Common Ingredient Profiles of Multi-Ingredient Pre-Workout Supplements. Nutrients, 11(2), 254. https://doi.org/10.3390/nu11020254
Page 2 Lines 59. “There is however evidence”. Please review the grammar of this sentence. Corrected
Page 2 Lines 66 to 68. If there is not empirical data “that” highlight a positive effect on physical performance, why is it controversial? Controversial is when some studies found effects and others did not find effects. Please the authors should rephrase this paragraph to justify the necessity of performing this study.
It was a mistake. Article text has been corrected.
Page 2 Lines 70 to 80. I do not understand the rationale of this paragraph that describes how the effects of citrulline are not found neither for untrained, moderately trained and professional athletes. This paragraph should be rewritten to describe with some evidence why it seems that citrulline does not have effects of all type of athletes. The main scientific question is if citrulline is effective in the professional sportsmen. As it is noted in the article, in previous studies when highly skilled subjects were examined, no positive impact on performance was observed
The introduction lacks of hypothesis. I think it would be very interesting to see what the authors predicted before performing the study. Do you think that citrulline was going to have certain effects? This study was a pilot by nature so we attempted to examine a scientific hypothesis that could be tested in a later trial.
Methods
Sample
Why is this study done in soccer players? Is there any special reason? Please justify why this sample.
According to non-published data citrulline often used by football players. Taking into account the absence of any published data with this specific group we decided to examine this in adult players of the elite level will have interest.
Procedure
Authors must explain and justify why they used this specific exercise and no other. Is this exercise representative of the real efforts of soccer? Has this exercise been used in previous studies? in which studies? Please describe this.
This exercise is typical for the top-level football players. It could be called ecological and it provides similar movement activity for all participants (it was confirmed during the study).
Results
Results are well presented
Discussion
The discussion needs English proof-reading and a more elaborated interpretation of the results.
From the line 290 to 326 the authors cite and explain multiple studies but they should have been create relationship between them and summarize their findings, rather than explaining all of them. Please review and rephrase considerably this part of the discussion. Corrected
An example of proof-reading is in Page 8 Line 269. Please rephrase the sentence “it´s found”, this is not a typical expression in academic and formal English. Corrected
